# Association between Corrected QT Interval and C-Reactive Protein in Patients with Inflammatory Bowel Diseases

**DOI:** 10.3390/medicina56080382

**Published:** 2020-07-30

**Authors:** Angelo Viscido, Annalisa Capannolo, Renata Petroni, Gianpiero Stefanelli, Giulia Zerboni, Massimo De Martinis, Stefano Necozione, Maria Penco, Giuseppe Frieri, Giovanni Latella, Silvio Romano

**Affiliations:** 1Gastroenterology Unit, Department of Life, Health and Environmental Sciences, University of L’Aquila, 67100 L’Aquila, Italy; annalisacap@tiscali.it (A.C.); giastefanelli@gmail.com (G.S.); g.frieri@libero.it (G.F.); giolatel@tin.it (G.L.); 2Cardiology Unit, Department of Life, Health and Environmental Sciences, University of L’Aquila, 67100 L’Aquila, Italy; renata.petroni@gmail.com (R.P.); maria.penco@univaq.it (M.P.); silvio.romano@univaq.it (S.R.); 3Di Lorenzo Clinic, Avezzano, 67100 L’Aquila, Italy; 4IBD Unit, S. Filippo Neri Hospital, 00135 Rome, Italy; giulia.zerboni@gmail.com; 5Clinical Immunology Unit, Department of Life, Health and Environmental Sciences, University of L’Aquila, 67100 L’Aquila, Italy; demartinis@cc.univaq.it; 6Statistics Unit, Department of Life, Health and Environmental Sciences, University of L’Aquila, 67100 L’Aquila, Italy; stefano.necozione@univaq.it

**Keywords:** inflammatory bowel diseases, ulcerative colitis, Crohn’s disease, ECG, QT interval, C-reactive protein

## Abstract

*Background and objectives:* Electrocardiograph abnormalities (i.e., QT interval prolongation) have been described in inflammatory bowel diseases (IBD). We aimed to measure the QT interval in a cohort of patients with IBD and to analyze its relationship with clinical and inflammatory activity. *Materials and Methods:* We performed a cross-sectional study that included 38 IBD outpatients and 38 “age- and sex-matched” healthy controls. Nine patients had active IBD, and 29 were in clinical remission. Among the latter, 10 patients had sustained (lasting >1 year) and 19 had short-term remission (≤1 year). Corrected QT (QTc) interval was measured on standard 12-lead electrocardiograph. A systematic review of the literature on studies investigating the QT interval in patients with IBD was also performed. *Results:* QTc interval values were similar between IBD patients and healthy controls (417.58 ± 22.05 ms vs. 409.13 ± 19.61 ms, respectively; *p*: 0.479). Patients with active IBD had significantly higher QTc values (435.11 ± 27.31 ms) than both controls (409.13 ± 19.61 ms) and patients in remission (412.14 ± 17.33 ms) (*p*: 0.031). Post hoc analysis showed that the difference in QTc values between active IBD and remission was attributable to the group of patients with sustained remission (*p* < 0.05). Lastly, a significant correlation between QTc interval and C-reactive protein (CRP) values was observed (Spearman test: r = 0.563; *p*: 0.0005). *Conclusions:* Our study demonstrates an association between QTc duration and both clinical and inflammatory activity in patients with IBD. The higher the CRP value, the longer is the QTc duration. For practical purposes, all patients with active IBD should undergo a standard ECG. Prescription of drugs able to modify the QT interval should be avoided in patients with active IBD. The systematic review of the literature indicated that this is the first published study demonstrating an association between the QTc duration and CRP values in patients with IBD.

## 1. Introduction

Inflammatory bowel diseases (IBDs), i.e., ulcerative colitis (UC) and Crohn’s disease (CD), are characterized by an idiopathic inflammatory process of the intestine and a chronic relapsing course [1,2]. Intestinal inflammation is responsible for signs, symptoms, and complications of IBD, being capable to trigger inflammatory responses even beyond the bowel. Inflammatory cells and mediators can flow from the intestine into the systemic circulation. The more the intestinal inflammation is active, the higher is the inflammatory overflow into circulation [1,2,3,4]. Systemic inflammation is responsible for many effects, of which the most known is the acute phase response [5]. C-reactive protein (CRP), the major acute-phase protein, is rapidly produced in the liver consequent to stimulation by the cytokines tumor-necrosis-factor-α (TNFα), interleukin-1 (IL-1), and interleukin-6 (IL-6) originating at the site of inflammation [6]. The acute phase CRP represents an accurate marker of inflammatory activity being able to detect subclinical active inflammation [7].

In recent years, systemic inflammation has been increasingly recognized as a factor crucially involved in modulating cardiac function [8,9]. The connection between inflammation and cardiovascular disease (CVD) has been extensively elucidated in patients with rheumatoid arthritis (RA) and other connective diseases [10,11]. Growing evidence suggests that, in patients with IBD, intestinal inflammation can cause cardiovascular morbidity, as well [12,13,14,15]. Epidemiological studies have demonstrated that patients with clinically active IBD have an increased risk of atrial fibrillation (AF), myocardial infarction (MI), and death due to CVD [16,17,18,19,20,21,22,23,24,25]. Notably, morbidity and mortality due to CVD are exclusively observed during active IBD flares. Studies based on electrocardiogram (ECG) findings indicate that patients with IBD are at a higher risk of developing QT interval abnormalities, in particular heart-rate-corrected QT interval (QTc) [26,27,28,29,30,31,32,33]. 

The QTc interval indicates the duration of the action potential in the ventricles, representing the sum of ventricular depolarization and repolarization. It is well established that the more QTc is prolonged, the higher the risk is that abnormal premature depolarization occurs prior to completion of repolarization. This can generate malignant ventricular arrhythmias, particularly torsade de pointes [34]. Torsade de pointes specifically develops in patients presenting with a marked QTc prolongation (>500 ms) at the ECG, and it can degenerate into ventricular fibrillation and sudden cardiac death. QTc prolongation may result from drugs or electrolyte disturbances interfering with cardiomyocyte electrophysiology. Other currently recognized causes of QT prolongation include heart diseases, endocrine and metabolic disorders, liver diseases, nervous system injuries, and infections [34,35].

Growing evidence indicates that the QTc interval can be modulated by systemic inflammation [34,35]. However, the relationship between systemic inflammation and QTc interval has not been elucidated in IBD.

The present case-control study was conducted to measure the QT interval in a cohort of patients with IBD free of CVD and to analyze its relationship with clinical and inflammatory activity, evaluated by means of standard clinical indices and acute phase CRP. A systematic review of the literature, aimed at identifying all the studies investigating the QT interval in patients with IBD, was also performed.

## 2. Materials and Methods

### 2.1. Study Population

This was a single-centre, observational, cross-sectional, case-control study. The study population consisted of consecutive outpatients with IBD admitted to our clinic between 1 December 2018 and 28 February 2019. The diagnosis of both ulcerative colitis (UC) and Crohn’s disease (CD) was based on standard clinical, radiological, endoscopic, and histological criteria [36].

Clinically active disease was defined as a partial Mayo score (PMS) ≥ 2 in patients with UC [37,38] and as a Harvey–Bradshaw index (HBI) ≥ 5 in patients with CD [39]. Clinical remission was sub-classified in (1) “sustained clinical remission”, i.e., lasting >1 year after a course of steroids, and (2) “short-term remission”, i.e., lasting ≤1 year after a relapse needing steroids. 

Inflammatory activity was defined as a CRP value > 0.5 mg/dL.

### 2.2. Exclusion Criteria


History of cardiovascular, metabolic, and endocrine diseases;Electrolyte disturbances;Assumption of drugs affecting QT interval (with the exception of those required to cure IBD).


The baseline characteristics, including demographic (age and gender) and clinical (UC or CD, disease duration, extent, clinical course, and prior and current medications) data, of each enrolled patient, were recorded. Ethics approval was obtained by the Internal Review Board of the University of L’Aquila (protocol number: 40/2018; 13 November 2018). Both patients and controls provided signed informed consent.

### 2.3. Control Group

The control group comprised age- (± 2 years) and sex-matched healthy subjects. All the healthy subjects were free of any medications and without electrolyte disturbances. 

### 2.4. ECG Analysis

Both IBD patients and controls underwent a standard 12-lead ECG (amplitude of 10 mm/mV; the paper speed of 25 mm/s) during a routine ambulatory visit.

The ECGs were uploaded to a computer, using a high-resolution optical scanner. Therefore, we performed on-screen manual measurements on the computer. Only the cycles with normal morphologic characterized beats were used; artifacts, ectopic, and post-extrasystolic complexes were excluded. All the patients had sinus rhythm. The QT interval was measured from the onset of the QRS complex to the end of the T-wave in lead II. Biphasic T-waves were measured to the time of the final return to the baseline. QTc intervals were calculated by using Bazett’s formula (QTc = QT interval/√RR interval) [40,41].

*Normal QTc interval*. QTc intervals of 450 milliseconds (ms) in adult men and 460 ms in women are generally considered as the upper normal limits [42]. We defined “borderline” QTc interval prolongation as the values over 450 ms in men and 460 ms in women. According to ESC guidelines, long QT syndrome was defined as the presence of a QTc ≥480 ms, for both men and women [43].

### 2.5. Statistical Analysis

The normality of the distribution of all the continuous variables was evaluated by the Shapiro–Wilk test. The comparison between the study groups was performed in two ways, depending on their number. The comparison between two groups was evaluated by Student’s *t*-test for independent data. Instead, the comparison between three groups was evaluated by one-way analysis of variance, and post hoc analysis was performed by means of Tukey’s HSD test. The correlation between QTc and CRP values was analyzed by Spearman’s rank correlation coefficient (rho).

Results are presented as mean ± standard deviation (SD). All the tests were two-tailed. All statistical analyses were performed by using SAS 9.4.

### 2.6. Systematic Review

Search strategy: we conducted a systematic literature search in order to identify studies investigating the QTc interval in patients with IBD. The electronic search was performed in PubMed, Scopus, Mendeley, and ResearchGate, up to June 2020, using the following mesh terms: inflammatory bowel diseases, ulcerative colitis, Crohn’s disease, ECG, and QT interval. In addition, the reference lists of the selected articles were reviewed, to identify additional relevant studies.

## 3. Results

### 3.1. Study Population and Baseline Assessment

A total of 38 patients with IBD were enrolled in the study. The baseline demographic and clinical characteristics of all the patients are shown in Table 1. In particular, 26 patients had UC, and 12 patients had CD. Nine patients (5 CD and 4 UC) had active disease, while twenty-nine patients were in clinical remission. Among the latter, 10 patients had sustained clinical remission and 19 patients had short-term remission.

### 3.2. Case-Control QTc Comparison 

The mean ages of IBD patients and matched controls were 46 ± 11 and 45 ± 11 years, respectively, with 52.6% (*n*: 20) being females and 47.4% (*n*: 18) being males in each group.

Neither patients nor the controls had long QT syndrome. The overall prevalence of “borderline” QTc interval prolongation was 7.9% (3 patients) in the IBD population, and zero in the control group. All the IBD patients with “borderline” QTc interval prolongation had active disease.

The mean QTc interval values were similar between IBD patients and healthy controls (417.58 ± 22.05 ms vs. 409.13 ± 19.61 ms, respectively; *p*: 0.479).

After stratification based on disease activity, a significant difference in mean QTc values was observed between patients with active disease and controls (435.11 ± 27.31 ms vs. 409.13 ± 19.61 ms, respectively; *p*: 0.014). Figure 1 reports the results of the post hoc comparison of QTc values between three groups: healthy controls, patients in remission, and patients with active IBD. QTc values of patients with active IBD are significantly higher in respect to both patients in remission and healthy controls (*p*: 0.031).

### 3.3. QTc and Disease Activity

Table 2 shows that patients with active disease had significantly higher mean CRP and QTc values in respect to those in clinical remission (*p*: 0.0045 and *p*: 0.0047, respectively).

Subsequently, we applied the sub-classification of clinical remission into “sustained” and “short-term” remission. Figure 2 shows the results of the post hoc comparison of mean QTc values in patients with active disease, those with short-term remission, and those with sustained remission. The mean QTc values were significantly different between patients with active disease and those with sustained remission (*p* < 0.05, post hoc analysis with HSD Tukey test after one-way ANOVA).

Figure 3 shows the correlation between QTc and CRP values. Spearman’s coefficient of rank correlation (rho) was 0.563 (*p*: 0.0005), indicating a good correlation.

There were no significant relationships between the QTc values and all the other baseline characteristics, i.e., gender, age, disease duration, type of IBD (UC or CD), and therapy.

### 3.4. Systematic Review

The search retrieved 267 articles. Figure 4 shows the PRISMA diagram summarizing the sequence of the literature selection. Seven studies reported on QTc interval [27,28,29,30,31,32,33]. Table 3 summarizes the results of the studies that measured the QTc interval in patients with IBD. Studies were not homogeneous with respect to the design, patients’ characteristics, and clinical activity. It is noteworthy that none of the studies evaluated the relationship between QTc and CRP.

## 4. Discussion

The results of our study demonstrate that both clinical and inflammatory activity are significantly associated with QTc duration in patients with IBD. The higher the value of CRP was, the longer the QTc duration was. This finding is in line with the observations made in other chronic inflammatory conditions, such as RA [10,11,34,35].

The need to evaluate cardiac function in patients with IBD became apparent in the last few years, following ever-increasing evidence that systemic inflammation may affect heart function [12,13,14,15]. Population-based studies demonstrate that disease activity in IBD is associated with an increased risk of morbidity and mortality due to CVD [16,17,18,19,20,21,22,23,24,25]. Furthermore, a number of observations indicate that patients with IBD are at a higher risk of developing ECG abnormalities, i.e., QTc interval [21,26,27,28,29,30,31,32,33]. QTc interval represents the most investigated ECG parameter in patients with IBD [27,28,29,30,31,32,33]. Studies identified by our systematic review are not homogeneous with respect to the design and the characteristics of the patients. Most of them indicate that patients with IBD are at a higher risk of developing QTc interval prolongation [27,29,30,31,33]. Two studies did not find differences in QTc values between IBD patients and controls [28,32]. The first of these two studies could have a methodological bias, as the QT-interval was measured by using a magnifying lens instead of computer-assisted calculations [28]. The other study could have a selection bias, as none of the patients had a clinically active disease [32]. Overall, a correlation with the inflammatory activity measured by means of CRP was lacking in all the studies (Table 3). However, a connection between the QTc interval at ECG and IBD activity is beyond any doubt.

The QT interval indicates the duration of the action potential in the ventricles, which represents the sum of ventricular depolarization and repolarization. An action potential is caused by the transmembrane flow of ions, including inward depolarizing currents, mainly through sodium and calcium channels, and outward repolarizing currents, mainly through potassium channels [34,35]. 

There is growing evidence that inflammatory cytokines have direct effects on cardiac ion channels, resulting in a prolongation of cardiomyocyte action potential duration, and thus on the QT interval on the surface ECG [44,45]. The term “inflammatory cardiac channelopathies” has been recently introduced to categorize these pathogenetic mechanisms [46,47,48,49]. Experimental models indicate that TNFα, IL-1, and IL-6 may affect cardiac electrophysiology by modulating specific ion (potassium and/or calcium) channels critically involved in action potential duration [44]. Perfused hearts from transgenic mice overexpressing TNFα have a prolonged action potential duration and ventricular arrhythmias associated with reduced expression of the potassium channel protein in the cardiomyocytes [50,51]. Studies on cultured cardiomyocytes derived from different animal models indicate that TNFα prolongs the action potential duration by modulation of potassium channel [52,53,54], whereas IL-1 and IL-6 cause action potential prolongation by modulation of both potassium and calcium channels [55,56,57,58].

Clinical studies including patients with RA demonstrated that cytokines may really affect QTc interval duration in humans. Increased concentrations of circulating TNFα, IL-1, and IL-6 are associated with QTc interval prolongation in patients with RA [59,60]. Moreover, increased levels of IL-6 are also associated with QTc prolongation in the presence of systemic inflammation deriving from other diseases besides RA, in particular immune-inflammatory and infective diseases [61]. Notably, patients with active RA who receive anti–IL-6 receptor monoclonal antibodies (tocilizumab) have a significant and rapid reduction of the QT interval values, which correlates with the decrease in both CRP and, more strongly, circulating TNFα levels [62,63].

In addition to the direct action on ion channels, systemic inflammation can also determine the prolongation of the action potential duration in an indirect manner, by inducing autonomic nervous system dysfunction [34]. Cytokines can cause sympathetic activation by targeting the autonomic centers of the brain, and such activation ultimately affects calcium and potassium conductance, leading to prolongation of cardiomyocyte action potential duration and QTc interval [34,35]. It has been observed that sympathetic activation correlates with clinical and inflammatory activity in patients with IBD [64].

It is noteworthy that systemic inflammation, in addition to causing functional myocardial abnormalities, can cause also structural heart abnormalities, e.g., accelerating coronary atherosclerosis [15,34,35,65]. Concurrency of action potential prolongation and preexisting structural myocardial abnormalities may explain the epidemiological observation that patients with active IBD have an increased risk of morbidity and mortality due to CVD [34,35]. Studies on atherosclerotic animal models of myocardial infarction demonstrate that the induction of systemic inflammation determines a significant prolongation of the action potential duration with a consequent increased incidence of fatal reentrant ventricular arrhythmias [66]. Notably, TNFα antagonism and IL-1 inhibition are able to normalize the action potential duration and to significantly reduce the risk of fatal ventricular arrhythmias in these types of animal model [67,68]. 

All the abovementioned observations, including results of our study, indicate that systemic inflammation has a pivotal role in increasing the risk of CVD in patients with active IBD. Further evidence in favor of a link between inflammation and CVD comes from the observation that treatment optimization could reverse the risk of CVD in patients with inflammatory diseases. A meta-analysis demonstrated that, in patients with RA, both anti-TNFα therapies and methotrexate are associated with a decreased risk of major cardiovascular events [69]. Notably, a recent prospective study including patients with RA demonstrated that intensive anti-inflammatory treatment resulting in a stable low-disease activity is associated with a decrease in deaths due to CVD [70].

Increased surveillance with treatment optimization aimed at reduction of inflammation should be warranted in patients with IBD, as well as with other inflammatory conditions [71]. It has been demonstrated that healing of inflammatory lesions is associated with better outcomes in IBD [72]. It could be hypothesized that sustained healing of lesions could prevent CVD, since our results show a significant effect on QTc duration only in patients with sustained remission. The availability of potent biologic therapy is contributing to determine a condition of sustained remission in an increasing number of patients [73,74]. Furthermore, in order to prevent CVD in IBD, care must be taken also to metabolic syndrome that can complicate intestinal inflammation [75,76,77].

Our study demonstrates, for the first time, an association between QTc duration and both clinical and inflammatory activity in patients with IBD. However, it presents, along with some strengths, some important limitations. Regarding the strengths, we studied a homogeneous cohort of IBD patients with no recognized causes of QTc prolongation, the controls were age- and sex-matched, and CRP was used as an objective marker of systemic inflammation. With respect to the limitations, the small number of patients did not allow us to accurately evaluate the role of some patients’ characteristics, i.e., gender, type of IBD (UC or CD), disease location/extension, and therapies. Furthermore, data on endoscopy and fecal calprotectin are lacking. Endoscopy and fecal calrptectin should be included in future studies on ECG alterations in patients with IBD.

In conclusion, the present study demonstrates that patients with active IBD have increased QTc duration and that QTc duration significantly correlates with CRP values. It remains to be clarified by means of prospective studies whether QTc may represent a marker of CVD risk in patients with active IBD.

For practical purposes, clinicians may want to consider that all the patients with active IBD should receive an ECG, and QT interval prolongation should be kept in mind when prescribing QT-prolonging drugs. IBD units must have the ability to take an ECG to all patients.

## Figures and Tables

**Figure 1 medicina-56-00382-f001:**
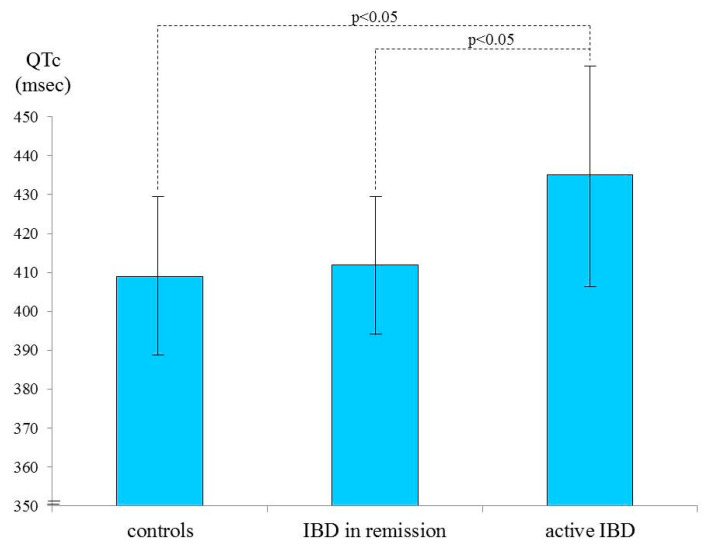
Post hoc comparison of QTc values between healthy controls (409.13 ± 19.61 ms), patients in remission (412.14 ± 17.33 ms), and patients with active IBD (435.11 ± 27.31 ms). QTc values of patients with active IBD are significantly higher in respect to both patients in remission and healthy controls (*p*: 0.031). QTc = corrected QT; IBD = inflammatory bowel diseases.

**Figure 2 medicina-56-00382-f002:**
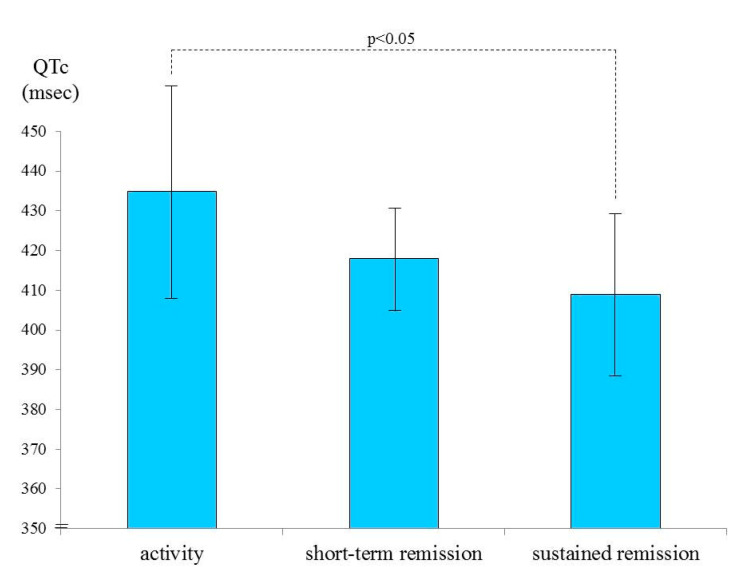
Post hoc comparison of QTc values in patients with active disease (435.11 ± 27.31 ms), those with short-term remission (418.10 ± 12.77 ms), and those with sustained remission (409.00 ± 18.85 ms). The mean QTc values were significantly different between active disease and sustained remission (*p* < 0.05).

**Figure 3 medicina-56-00382-f003:**
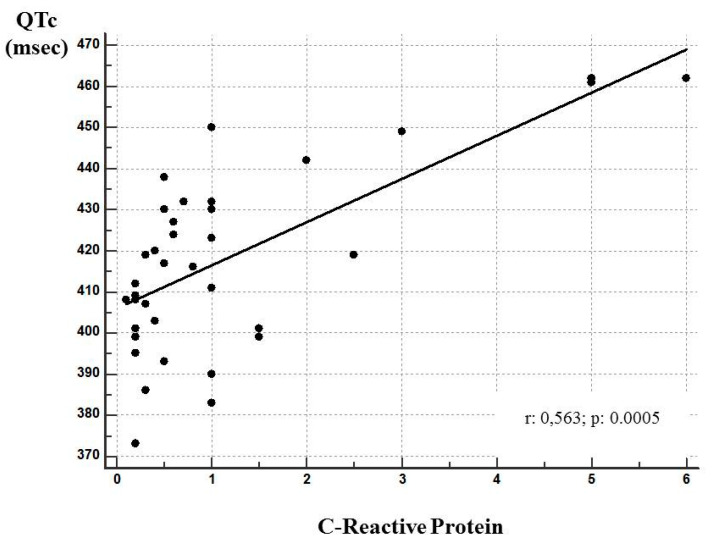
Correlation between QTc interval and CRP (mg/dL) values. The correlation is significant (Spearman test: rho = 0.563; *p* < 0.0005).

**Figure 4 medicina-56-00382-f004:**
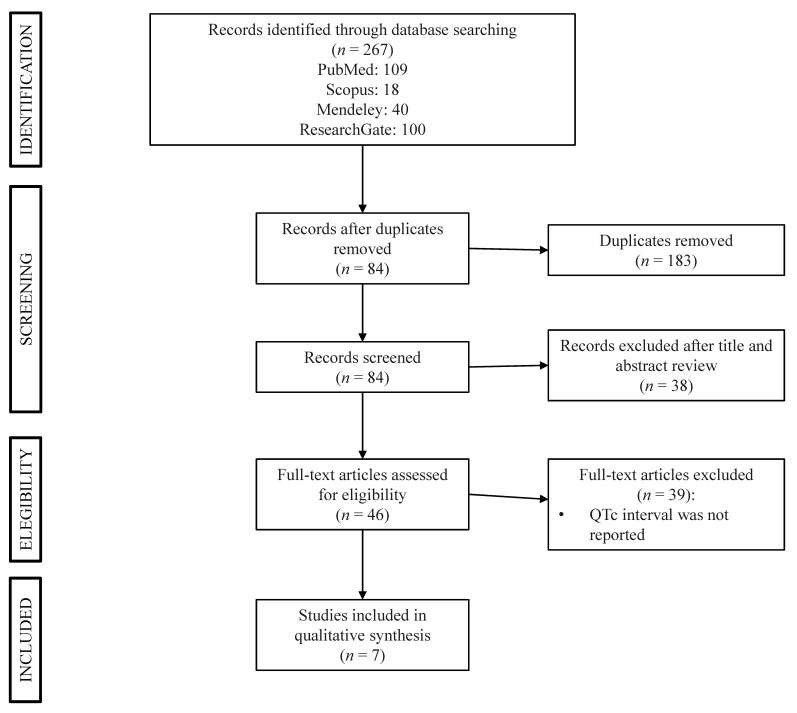
PRISMA flow diagram of the systematic literature search.

**Table 1 medicina-56-00382-t001:** Baseline demographic and clinical characteristics of the cohort of patients with inflammatory bowel disease (IBD) and controls. Remission was categorized as sustained (>1 year) or short-term (⩽1/year).

Characteristic	N (%)
Gender (female/male)	20/18
Age (mean, years ± SD)	46 ± 11
Disease duration (mean, years ± SD)	12 ± 8
Ulcerative colitis	26 (68.4)
*Pancolitis*	*10 (38.5)*
*Left-colitis*	*13 (50.0)*
*Proctitis*	*3 (11.5)*
Crohn’s disease	12 (31.6)
*Ileo-colonic*	*9 (75.0)*
*Ileal*	*3 (25.0)*
Active disease	9 (23.7)
Remission	29 (76.3)
*Sustained*	*19 (65.5)*
*Short-term*	*10 (34.5)*
Therapy ^	
Mesalazine	29 (76.3)
Steroids	6 (15.8)
Thiopurines	4 (10.5)
Biologics	9 (23.7) *
CRP (mean, mg/dL ± SD)	1.11 ± 1.41
QTc (mean, ms ± SD)	417.579 ± 22.055

^ More than one medication was administered to some patients. * Biologics: infliximab in three patients, adalimumab in three patients, and vedolizumab in two patients. CRP = C-reactive protein; QTc = corrected QT.

**Table 2 medicina-56-00382-t002:** Mean CRP and QTc values in patients with clinically active IBD and in patients in clinical remission.

	Active IBD (9 pts)	Remission (29 pts)	*p*
CRP (mean, mg/dL ± SD)	3.000 ± 1.888	0.524 ± 1.887	0.0045
QTc (mean, ms ± SD)	435.111 ± 27.306	412.138 ± 17.328	0.0047

Notes: pts = patients.

**Table 3 medicina-56-00382-t003:** Studies evaluating QTc interval in patients with IBD (retrieved by a systematic review of the literature). Inflammatory activity measured by means of CRP was not evaluated in any of the studies.

Study	IBD Pts/Controls N°	QTc Findings	Clinical Activity
Curione, 2010 [27]	20/18 (no comorbidities)	Mean QTc significantly higher in IBD vs. controls	Not evaluated
Dogan, 2011 [28]	69/38 (no comorbidities)	Mean QTc similar between IBD and controls	Active disease in all patients
Yorulmaz, 2013 [29]	104/47 (no comorbidities)	QTc dispersion significantly higher in IBD vs. controls	Not specified
Pattanshetty, 2016 [30]	142/- (>50% with comorbidities)	Prolonged QTc interval in 46.5% of IBD patients	Not evaluated
Bornaun, 2017 [31]	36/36 (pediatric)	Mean QTc min significantly different between IBD and controls	Clinical remission in all patients
Acar, 2019 [32]	100/100 (no comorbidities)	Mean QTc similar between IBD and controls	Clinical remission in all patients
Erolu, 2020 [33]	25/20 (pediatric)	QTc dispersion higher in IBD vs. controls (especially in UC)	Clinical remission in all patients

Note: UC = ulcerative colitis. N° = number.

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
