# Peer review of "Association between Corrected QT Interval and C-Reactive Protein in Patients with Inflammatory Bowel Diseases"

_medicina, 2020, doi:10.3390/medicina56080382_

Round 1
Reviewer 1 Report
I have read with great interest the manuscript entitled "Association Between Corrected QT Interval and C-Reactive Protein in Patients with Inflammatory Bowel Diseases".
The paper describes the association between QTc and disease activity in patients with IBD. Statistics is well conducted and the message is of interest for clinicians.
The authors should point out in the discussion section the strengths and limitations of the study. Also, if possible, an analysis adjusting to IBD-related drugs should be done.
When reporting results, authors should review if measure unit for CRP is mg/dL or mg/L.
Author Response
Thanks a lot to the reviewer as I think that the quality of the paper has improved following his suggestions.
Point-by-point reply to the Reviewer.
- The authors should point out in the discussion section the strengths and limitations of the study.
We added the strengths and limitations of the study in the discussion section at page 13, lines 292-300.
- Also, if possible, an analysis adjusting to IBD-related drugs should be done.
The analysis adjusting to IBD-related therapy did not showed any significant association, probably due to the small number of patients. This was added in the results section at page 9, lines 187-188.
- When reporting results, authors should review if measure unit for CRP is mg/dL or mg/L.
We added the reference value of C-reactive protein at page 5, line 105.
Reviewer 2 Report
In this original study, the authors demonstrated a positive association between QTc duration and clinical as well as biological inflammatory activity in patients with IBD. Their conclusion has practical importance, suggesting that all patients with active IBD should undergo a standard ECG and, also, more importantly, prescription of medication able to modify the QT interval should be avoided in patients with active IBD. I appreciate the novelty of the study and the clarity of the manuscript. The article is easily to be read.
Other comments/suggestions:
- Abstract: 1. Material and Methods: From what I understand, this was a prospective study. Then, this should be mentioned, as it is more accurate than a retrospective one. 2. A systematic review of the literature on studies investigating the QT interval in patients with IBD was also performed, but there is nothing about this in the Conclusion. Please insert a brief sentence in the Conclusion or emphasize that this is the first study of its kind, after appropriate search of the literature.
- Introduction: very well written, concise and clear.
- Material and methods: 1. Please insert that this was a prospective study. 2. Please mention the period the study was carried out, as it is not written. 3. The authors mentioned “The diagnosis of both UC and CD was based on standard clinical, radiological, endoscopic and histological criteria”. Please insert reference, in order to be clear what criteria were used for diagnosis. 4. Please insert methods regarding C - reactive protein and normal values considered/defined as reference in your study, since they differ among laboratories. 5. Systematic review: Other databases should have been searched as well, not only Pubmed. I agree PubMed is good, but it is not quite relevant for a systematic review. Please include searches on other databases as well (EMBASE, Web of Science, Google scholar; Mendeley, Research Gate etc). Please review.
- Results: 1. Study population and baseline assessment: 1a. There were only 12 patients with CD. Systemic inflammation is considered to be more important in CD vs UC. This should be a limitation of the study. 1b. Nine patients with active disease vs those with inactive disease represent quite a small number. This also should be a limitation of the study. 1c. What type of disease did those 9 patients have? CD or UC? It should be mentioned, as these two types are quite different. 1d. From Table 1: Was there any sub-analysis regarding gender performed? 1e. From Table 1: Was there any sub-analysis regarding the E types of ulcerative colitis (extension) performed? 1f. Was there any sub-analysis regarding the L types of Crohn’s disease (location) performed? Was there any reason to not include patients with colonic disease only (L2)? 1g. Table 1: Please state, regarding medication, that more than 1 medication was administered in some patients. 1h. Normal values of CRP should have been mentioned before, otherwise they cannot be interpreted. 2. QTc and disease activity: Table 2: The values regarding QTc are also shown in Figure 1. 3. Systematic review: 3a. Lines 208-209, regarding Table 3: The authors mentioned that “It is noteworthy that none of the studies evaluated the relationship between QTc and CRP”. This sentence could be shortened and added to the title of the Table 3. If so, the last column of Table 4 can be deleted. 3b. As I mentioned before, other databases should have been searched. Then, more studies would have been retrieved. As an example: “Erolu E, Polat E. Cardiac Repolarization Properties in Children with Inflammatory Bowel Disease. Cyprus J Med Sci 2020; 5(2): 126-30”. Table 3 should be corrected, to include all studies that were published in the most common databases.
- Discussion: 1. Discussion should start with the findings of the study, instead of generalities, which were already mentioned. 2. This paragraph contains many aspects that were already presented in the Introduction. They should be removed. Other sentences refer to possible hypotheses regarding prolongation of the QT interval. Data from the literature are interesting, but they should be either written in the Introduction or linked to the findings of this study. 3. More in depth discussion should be made about the results of this study and the other findings in the literature. This is totally missing. 4. There are no comments/interpretations of the studies that were presented in the Table 3 (from the systematic review). They should be inserted, in comparison with the findings of the present study. 5. Strength of this study should be emphasized: first study in IBD analysing correlation between prolonged QTc interval and CRP and finding a moderate correlation between active disease and longer QTc. 6. Limitations of this study (above mentioned) should be presented. 7. As direction for further research, it should be included that studies analysing correlation between fecal calprotectine (surrogate of mucosal healing, much more accurate marker of inflammation than C-reactive protein) should be performed. 8. Also, conclusion of this study could be reformulated, in a different positive way (when access to ECG is limited): Since higher CRP is correlated with prolonged QTc, prescription of medication able to modify the QT interval should be avoided in patients with active IBD, in whom ECG cannot be performed, especially during this pandemic period. This would be a very useful plus!
Author Response
Thanks a lot to the reviewer as I think that the quality of the paper has greatly improved following his suggestions.
Point-by-point reply to the Reviewer.
Reviewer 2 comments:
- Abstract:
- Material and Methods: From what I understand, this was a prospective study. Then, this should be mentioned, as it is more accurate than a retrospective one.
This was a cross-sectional study. It was mentioned both in the abstract (page 2, line 30), and in the methods section (page 5, line 97).
- A systematic review of the literature on studies investigating the QT interval in patients with IBD was also performed, but there is nothing about this in the Conclusion. Please insert a brief sentence in the Conclusion or emphasize that this is the first study of its kind, after appropriate search of the literature.
We added a brief sentence at the end of the conclusion (page 2, line 47). The literature search was updated according to your suggestion.
- Introduction: very well written, concise and clear.
Thank you very much.
- Material and methods:
- Please insert that this was a prospective study.
The design of the study was cross-sectional. This was added at page 5 (line 97). I apologize for the confusion we have created with the previous version.
- Please mention the period the study was carried out, as it is not written.
We added the study period (page 5, line 98).
- The authors mentioned “The diagnosis of both UC and CD was based on standard clinical, radiological, endoscopic and histological criteria”. Please insert reference, in order to be clear what criteria were used for diagnosis.
The reference of the diagnostic criteria was added (reference 36).
- Please insert methods regarding C - reactive protein and normal values considered/defined as reference in your study, since they differ among laboratories.
We added the reference value of C-reactive protein at page 5, line 105.
- Systematic review: Other databases should have been searched as well, not only Pubmed. I agree PubMed is good, but it is not quite relevant for a systematic review. Please include searches on other databases as well (EMBASE, Web of Science, Google scholar; Mendeley, Research Gate etc). Please review.
According to your precious suggestion, we did the review again (page 7, line 148). A significantly higher number of paper has been found, and a figure (figure 4) was added to illustrate the sequence of the literature selection.
- Results:
- Study population and baseline assessment:
1a. There were only 12 patients with CD. Systemic inflammation is considered to be more important in CD vs UC. This should be a limitation of the study.
This is an important limitation of our study and we underlined this aspect in the discussion (page 13, line 294).
1b. Nine patients with active disease vs those with inactive disease represent quite a small number. This also should be a limitation of the study.
This is another limitation underlined in the discussion (page 13, line 294).
1c. What type of disease did those 9 patients have? CD or UC? It should be mentioned, as these two types are quite different.
Out of the 9 patients with active disease, 5 had CD and 4 had UC. These numbers were added at page 8, line 158.
1d. From Table 1: Was there any sub-analysis regarding gender performed?
1e. From Table 1: Was there any sub-analysis regarding the E types of ulcerative colitis (extension) performed?
1f. Was there any sub-analysis regarding the L types of Crohn’s disease (location) performed? Was there any reason to not include patients with colonic disease only (L2)?
The sub-analysis according to gender, age, disease duration, type of IBD (UC or CD), and therapy did not showed any association with the QTc values. We added this information at page 9, line 187. In the discussion section (page 13, line 294) we specified that the small number of patients could be responsible for this result.
Patients with colonic Crohn’s disease are lacking by chance. We did not visited patients with colonic CD during the study period.
1g. Table 1: Please state, regarding medication, that more than 1 medication was administered in some patients.
We added this information to Table 1.
1h. Normal values of CRP should have been mentioned before, otherwise they cannot be interpreted.
We added the reference value of C-reactive protein at page 5, line 105.
- QTc and disease activity: Table 2: The values regarding QTc are also shown in Figure 1.
Figure 1 illustrates the values in comparison with controls. Table 2 reports values of CRP and QTc only in patients, those with active disease and those in remission. It is actually a partial repetition, but we would like to keep both illustration as they focuses on different aspects of the study.
- Systematic review:
- 3a. Lines 208-209, regarding Table 3: The authors mentioned that “It is noteworthy that none of the studies evaluated the relationship between QTc and CRP”. This sentence could be shortened and added to the title of the Table 3. If so, the last column of Table 4 can be deleted.
According to your suggestion, the sentence was added to table 3, and the last column was deleted.
3b. As I mentioned before, other databases should have been searched. Then, more studies would have been retrieved. As an example: “Erolu E, Polat E. Cardiac Repolarization Properties in Children with Inflammatory Bowel Disease. Cyprus J Med Sci 2020; 5(2): 126-30”. Table 3 should be corrected, to include all studies that were published in the most common databases.
According to your suggestion, we did the review again (page 9, line 190). The new search allowed to identify 3 more papers on QTc in patients with IBD (respect to the previous version). Figure 4 was added to illustrate the sequence of the literature selection.
- Discussion:
- Discussion should start with the findings of the study, instead of generalities, which were already mentioned.
According to your suggestion, the discussion starts with the results, and generalities were removed.
- This paragraph contains many aspects that were already presented in the Introduction. They should be removed. Other sentences refer to possible hypotheses regarding prolongation of the QT interval. Data from the literature are interesting, but they should be either written in the Introduction or linked to the findings of this study.
Aspects already presented in the introduction were removed. Data from the literature were linked to the findings of this study (page 10, lines 219-233).
- More in depth discussion should be made about the results of this study and the other findings in the literature. This is totally missing.
We discussed our results respect to the results of the other studies retrieved by the systematic review at page 10 (lines 224-233).
- There are no comments/interpretations of the studies that were presented in the Table 3 (from the systematic review). They should be inserted, in comparison with the findings of the present study.
As specified in the previous point, we discussed our results respect to the results of the other studies retrieved by the systematic review at page 10 (lines 224-233).
- Strength of this study should be emphasized: first study in IBD analysing correlation between prolonged QTc interval and CRP and finding a moderate correlation between active disease and longer QTc.
Strengths of the study were presented in the discussion at page 13, lines 292-296.
- Limitations of this study (above mentioned) should be presented.
Limitations of the study were presented in the discussion at page 13, lines 296-299.
- As direction for further research, it should be included that studies analysing correlation between fecal calprotectine (surrogate of mucosal healing, much more accurate marker of inflammation than C-reactive protein) should be performed.
The need of analyze correlation with endoscopy and fecal calprotectin has been added at page 13, line 299.
- Also, conclusion of this study could be reformulated, in a different positive way (when access to ECG is limited): Since higher CRP is correlated with prolonged QTc, prescription of medication able to modify the QT interval should be avoided in patients with active IBD, in whom ECG cannot be performed, especially during this pandemic period. This would be a very useful plus!
We added the statement that IBD units must have the ability to take an ECG to all patients (page 13, line 307). We did not refer to the pandemic as our scientific society (IGIBD) stated that telemedicine should be reserved only to patients that need no therapeutic changes, while a visit should be guaranteed to all patients with active disease.
Round 2
Reviewer 2 Report
I am very pleased with the way the manuscript appears now. The authors considered my suggestions and agreed with them. The modifications were made accordingly. It is true, they had to perform more work for this new version, but the manuscript is now complete. It was worth it! Congratulations to the authors!
Only please correct a typo: “Calrptectin” - page 13, line 299